# In Situ Compatibilized Blends of PLA/PCL/CAB Melt-Blown Films with High Elongation: Investigation of Miscibility, Morphology, Crystallinity and Modelling

**DOI:** 10.3390/polym15020303

**Published:** 2023-01-06

**Authors:** Nantaprapa Tuancharoensri, Gareth M. Ross, Arisa Kongprayoon, Sararat Mahasaranon, Supatra Pratumshat, Jarupa Viyoch, Narin Petrot, Wuttipong Ruanthong, Winita Punyodom, Paul D. Topham, Brian J. Tighe, Sukunya Ross

**Affiliations:** 1Biopolymer Group, Department of Chemistry, Faculty of Science, Naresuan University, Phitsanulok 65000, Thailand; 2Center of Excellence in Biomaterials, Faculty of Science, Naresuan University, Phitsanulok 65000, Thailand; 3Department of Pharmaceutical Technology, Faculty of Pharmaceutical Sciences, Naresuan University, Phitsanulok 65000, Thailand; 4Center of Excellence for Innovation in Chemistry, Naresuan University, Phitsanulok 65000, Thailand; 5Department of Mathematics, Faculty of Science, Naresuan University, Phitsanulok 65000, Thailand; 6Center of Excellence in Nonlinear Analysis and Optimization, Faculty of Science, Naresuan University, Phitsanulok 65000, Thailand; 7Department of Computer Science and Information Technology, Faculty of Science, Naresuan University, Phitsanulok 65000, Thailand; 8Center of Excellence in Materials Science and Technology, Chiang Mai University, Chiang Mai 50200, Thailand; 9Department of Chemistry, Faculty of Science, Chiang Mai University, Chiang Mai 50200, Thailand; 10Aston Institute of Materials Research, Aston University, Birmingham B4 7ET, UK

**Keywords:** elongation, blend, poly(lactic acid), compatibilizer, films

## Abstract

Ternary-blended, melt-blown films of polylactide (PLA), polycaprolactone (PCL) and cellulose acetate butyrate (CAB) were prepared from preliminary miscibility data using a rapid screening method and optical ternary phase diagram (presented as clear, translucent, and opaque regions) as a guide for the composition selection. The compositions that provided optically clear regions were selected for melt blending. The ternary (PLA/PCL/CAB) blends were first melt-extruded and then melt-blown to form films and characterized for their tensile properties, tensile fractured-surface morphology, miscibility, crystallinity, molecular weight and chemical structure. The results showed that the tensile elongation at the break (%elongation) of the ternary-blended, melt-blown films (85/5/10, 75/10/15, 60/15/25 of PLA/PCL/CAB) was substantially higher (>350%) than pure PLA (ca. 20%). The range of compositions in which a significant increase in %elongation was observed at 55–85% *w*/*w* PLA, 5–20% *w*/*w* PCL and 10–25% *w*/*w* CAB. Films with high %elongation all showed good interfacial interactions between the dispersed phase (PCL and CAB) and matrix (PLA) in FE-SEM and showed improvements in miscibility (higher intermolecular interaction and mixing) and a decrease in the glass transition temperature, when compared to the low %elongation films. The decrease in *M_w_* and *M_n_* and the formation of the new NMR peaks (^1^H NMR at 3.68–3.73 ppm and ^13^C NMR at 58.54 ppm) were observed in only the high %elongation films. These are expected to be in situ compatibilizers that are generated during the melt processing, mostly by chain scission. In addition, mathematical modelling was used to study the optimal ratio and cost-effectiveness of blends with optimised mechanical properties. These ternary-blended, melt-blown films have the potential for use in both packaging and medical devices with excellent mechanical performance as well as inherent economic and environmental capabilities.

## 1. Introduction

The development of sustainable and natural polymers-based resources as viable alternatives to fossil-based plastics has attracted much interest for many applications [1,2,3,4,5,6]. Polyester, such as poly(l-lactide) (PLLA), polycaprolactone (PCL), polyglycolide (PGA), and polybutylene succinate (PBS), are a versatile biodegradable family of sustainable polymers used in many applications such as food packaging, agriculture, drug delivery and medical devices [1,7,8,9]. There is now a growing interest in biodegradable blends with high elongation and resultant impact strength [10]. Poly(l-lactide) (PLLA), generally termed “PLA”, is currently one of the most widely used sustainable polymers that exhibits both biodegradability and tissue compatibility. Materials containing PLA have been designed and fabricated in various forms, such as nanogels, films, nanofibers, and 3D printing [11,12,13,14]. PLA is a brittle plastic with high mechanical strength; therefore, many researchers have investigated ways to improve its mechanical properties, especially its ductility or %elongation [15,16,17].

Blending PLA with other polymers is favored among various modification methods (such as synthesis of copolymers of PLA) to improve the properties of PLA, especially its brittleness, because of its cost-effectiveness at the industrial scale [18]. Examples of polymers used to blend with PLA are: Polycaprolactone (PCL) [17,19], thermoplastic polyurethane (TPU) [20], poly(ethylene-*co*-vinyl alcohol) [21], poly(butylene adipate-*co*-terephthalate) (PBAT) [22], and cellulose acetate butyrate (CAB) [23]. For example, in PLA/PCL reactive blends, PCL-g-PLA copolymers are generated and act as a compatibilizer, resulting in improved miscibility between the PCL and PLA phases [24]. Compatibilizers can be added or formed in situ to enhance interfacial adhesion between polymer phases that can result in improved mechanical properties [25,26,27,28]. In the literature, the compatibility of blends have been reported by the incorporation of catalysts (such as Ti(OBu)_4_, Sn(Oct)_2_, zinc borate, zinc acetate, titanium pigment, tetrabutyl titanate) as well as through the generation of transesterification in melt blends [21,22,29]. Even though there are many research works on blends of PLA, the enhancement in tensile elongation of PLA is still a challenge.

In this paper, PLA was chosen for melt-blending with two other biodegradable polymers (PCL and CAB) as a ternary blend, without the additional incorporation of plasticizers or copolymers, with the aim of reducing the brittleness of PLA based films. An important point to note is that binary blends are noticeably exemplified in the literature; however, ternary blends appear significantly less frequent and are not yet fully understood. In this work the goal was not simply to find three components that are completely miscible with each other, but rather to explore the structural effects of partial miscibility that can led to the enhancement of ductility of PLA. PCL was selected due to its flexible chains that can improve the brittleness of PLA, but it is known to be immiscible with PLA; therefore, CAB was chosen as the third component because of its known compatibility with ester-containing polymers [17,30].

Generally, the work on polymer blends aims to exploit certain compositions that are expected to enhance the mechanical properties of the blends. Appropriated compositions for melt-blending of PLA/PCL/CAB were selected using guidance from the rapid screening method represented by an optical ternary phase diagram [30]. Our previous work discovered that solution blending gives a direct prediction of apparently miscible ternary compositions in melt blends. The selected compositions were first melt-extruded before being melt-blown into the form of films, which were termed “ternary-blended, melt-blown films”. For the first time, remarkably high values of %elongation of up to 700% (avg. 350%) were observed in these blends that had no additional copolymers or compatibilisers. Ternary-blended, melt-blown films of PLA/PCL/CAB were characterized for their properties such as tensile strength, tensile elongation, tensile-fractured surface morphology, miscibility, thermal properties, molecular weight, crystallinity, chemical functionality and chemical structure. In addition, mathematic modelling was used to study the optimal ratio and cost-effectiveness for use of this ternary-blended, melt-blown film in industry.

## 2. Experimental

### 2.1. Materials

Poly(l-lactide) (PLLA) was supplied by Nature Works LLC, Plymouth, MN, USA Ingeo^TM^ (*M_w_* ≈ 100,000 gmol^−1^, 4043D, film grade). In this manuscript, PLA signifies the Nature Works product (PLLA). Polycaprolactone (PCL) (*M_w_* ≈ 15,000 gmol^−1^) was from Shenzhen Esun Industrial Co., Ltd., Shenzhen, China. Cellulose acetate butyrate (CAB) (*M_w_* ≈ 77,000 gmol^−1^, 2 wt% acetyl and 52 wt% butyryl content) was purchased from Shanghai Runwu Chemical Technology Co. Ltd., Shanghai, China. Chloroform (CF) was from RCI Labscan Limited, Bangkok, Thailand, used as solvent for solvent blending.

### 2.2. Miscibility Prediction

The miscibility between polymers (PLA, PCL and CAB) can be estimated by the critical solubility parameter difference, (Δδ)°^Crit^, from Coleman and Painter’s approach [31]. An estimate of the solubility parameter (δ) is obtained by dividing the sum of the molar attraction constants (F_i_) by molar volume (V) of the repeat units present in the polymer (Equation (1)). In our previous work [30], the difference in solubility parameter of polymer pairs were calculated and adjusted for their miscibilities following the (Δδ)°^Crit^, which depends on the interaction type between polymer pairs (non-polar and weak, moderate or strong polar interactions) (see Table 1).
(1)$$\delta =\sum \frac{F_{i}}{V}$$

### 2.3. Optical Ternary Phase Diagram of PLA/PCL/CAB Using Rapid Screening Method

Binary (PLA/PCL, PLA/CAB and PCL/CAB) and ternary (PLA/PCL/CAB) blends were prepared using the rapid screening method [19,30]. This rapid screening method allows many compositions to be constructed using apparent miscibility as the criteria for selection, which relies on the critical solubility parameter between polymer pairs [17,19,30,31]. This combinatorial technique uses transmission spectrophotometry and a multi-wavelength plate reader, which enables many samples to be measured quickly. Consequently, one can rapidly produce optical ternary phase diagrams in order to predict apparent miscibility using the optical clarity of solvent-blended films. Briefly, PLA, PCL and CAB polymers were dissolved in chloroform at 10 wt% and then pipetted into a 96-well plate at different polymer compositions. The solvent in the sample was evaporated slowly for 24 h to allow equilibrium morphology. Transmittance (%T) of the solvent-blended films at a wavelength of 450 nm was measured using a microplate reader. Solvent-blended films are defined as clear (apparently miscible) when %T ≥ 76, translucent at %T = 51–75, semi-translucent at %T = 31–50 and opaque (immiscible) at %T = 0–30.

### 2.4. Melt Processing of Binary and Ternary Blended Films

Compositions of PLA/PCL, PLA/CAB binary blends and PLA/PCL/CAB ternary blends for melt blending were selected form the solvent blended optical ternary phase diagram. These compositions were melt-blended with a twin screw extruder (LABTECH Model LTE16-40, Samutprakran, Thailand) to produce pellets using a plastic cutting machine. Conditions of the twin screw extruder were a screw speed of 100 rpm and temperatures of 150 °C at the feeding zone to 180 °C at the die. The pellets were then fabricated into form of films using a blow film extruder (LABTECH Model LE20–30/C & LF–250, Samutprakran, Thailand). The single screw extruder zone temperatures were set to 150 °C, 170 °C, 180 °C and 180 °C respectively. The screw speed and the film roll speed were 80 rpm and 400–500 rpm, respectively, to produce a controlled film thickness of 40 μm and film width of 18 cm.

### 2.5. Characterization of Ternary Melt-Blended Films

#### 2.5.1. Mechanical Properties

The mechanical properties of the blended films were observed using a universal tensile testing machine (INSTRON^®^ CALIBRATION LAB, Model 5965, Hopkinton, MA, USA), according to ASTM D638 standard test for tensile properties of plastics. The blended film samples were cut to 13 mm × 57 mm × 0.04 mm (5 pieces per sample). The testing conditions used a load cell of 1 KN and extension rate of 20 mm/min. The tensile strength at break, modulus at break and percentage of elongation at the break of blended films were reported.

#### 2.5.2. Morphology

The tensile-fractured surface morphology of ternary blend films was observed using Field-Emission Scanning Electron Microscopy (FE-SEM, ThermoFisher, Apreo S model, Hopkinton, MA, USA) in high vacuum mode.

#### 2.5.3. Miscibility

The miscibility of blended film samples was observed using attenuated total reflection Fourier transform infrared (ATR-FTIR) spectroscopy with a PerkinElmer Spectrum GX, Hopkinton, MA, USA (400–4000 cm^−1^).

#### 2.5.4. Crystallinity

Crystallinity of blended film samples was investigated by X-ray diffraction (XRD, Philips Model X’Pert Por, Eindhoven, The Netherland) with a diffraction angle range (2𝜃) from 5 to 80 degrees (Cu Kα, 1.54 Å).

#### 2.5.5. Thermal Analysis

The thermal properties of the ternary-blended, melt-blown films were investigated using Differential Scanning Colorimetry (DSC, Mettler model DAC1, Greifensee, Switzerland). Films were first heated from 25 to 200 °C, then cooled from 200 to −80 °C and a second heating cycle of −80 to 200 °C, at a heating and cooling rate of 10 °C/min under a nitrogen atmosphere.

#### 2.5.6. ^1^H and ^13^C NMR Analysis

Nuclear magnetic resonance (NMR), model Bruker Advance 400, Hopkinton, MA, USA, was used to study the chemical structures of ternary blend film. Briefly, different compositions of ternary blend film samples were dissolved in chloroform-D (CDCl_3_). After that, it was taken to be characterized by proton (^1^H) and carbon (^13^C) NMR.

#### 2.5.7. GPC Analysis

The gel permeation chromatography (GPC) was used to study the molecular weight of ternary-blended, melt-blown films. GPC analysis was carried out using a Waters 2414 refractive index (RI) detector, Hopkinton, MA, USA, equipped with Styragel HR5E 7.8 × 300 mm column (molecular weight resolving range = 2000–4,000,000). The different compositions of ternary melt-blended samples were dissolved in distilled tetrahydrofuran (THF) at a concentration of 3.33 mg/mL. THF was eluted at a rate of 1.0 mL/min at 40 °C and calibrated with polystyrene standards.

### 2.6. Modelling

The mathematical models were used to predict the optimal ratio of PLA, PCL, and CAB and the cost-effectiveness. The optimal ratio for the compound means to get the best composition to achieve the greatest %elongation without concerning the cost of raw materials. The model mainly focused on the ideal %CAB ratio and the relationship to the required %PLA ratio. Notice that %PCL is absent because it can be calculated easily later when %CAB and %PLA are already known. The detailed methodologies are described in Section 3.8.

## 3. Results and Discussion

### 3.1. The Selection of Compositions for Ternary-Blended, Melt-Blown Films

The preliminary predictions of miscibility between polymers pairs (PLA/PCL, PLA/CAB and PCL/CAB) were studied using the Coleman and Painter approach, following previous work [19,30]. Table 1 shows that PLA/PCL is predicted to be an immiscible blend that shows critical solubility parameter differences (∆δ) of 1.8 (cal cm^−3^)^1/2^, while PLA/CAB is predicted to be a miscible blend (∆δ = 0.1 (cal cm^−3^)^1/2^). PCL/CAB blends may be miscible or immiscible depending on the molecular interactions (∆δ = 1.7 (cal cm^−3^)^1/2^). The binary and ternary solvent-mediated blends based on PLA were constructed in the form of an optical ternary phase diagram (Figure 1a). Compositions from different regions (opaque, translucent and clear) were selected for melt-blending.

From melt processing experiments, the compositions from the opaque and translucent regions were not able to be processed due to high loading of PCL (>30% *w*/*w*), which results in high melt flow rates under the processing conditions of 150–180 °C (much higher than the melting temperature of PCL, T_m_ = 60 °C). The successful compositions (both binary and ternary blends) used for melt-blown films were in the clear region with a PLA content of at least 50% *w*/*w* (Figure 1b). In Figure 1b, the %elongation of these blends were classified in ranges; <50% (low %elongation, termed LE), 51–200%, 201–350% and >351% (high %elongation, termed HE). The range of compositions in which a significant increase in %elongation is: 55–85% *w*/*w* PLA, 5–20% *w*/*w* PCL and 10–25% *w*/*w* CAB. However, we were also concerned with the raw material cost per kilogram for the ternary blends, as the price is that CAB (105 USD/kg) > PCL (10 USD/kg) > PLA (4.5 USD/kg). Therefore, higher loading of PLA is more desirable in these ternary blends (further discussion in Section 3.8). Three samples of LE (red circles) and three samples of HE (blue squares) from Figure 1b and the binary-blended, melt blown films were chosen for detailed property characterization and further discussion.

### 3.2. Tensile Strength and %Elongation at Break

The tensile properties of “LE and HE” ternary-blended, melt-blown films (including binary-blended, melt-blown films) were studied (Figure 2). The results showed that pure PLA and binary blends (PLA/PCL and PLA/CAB) have low %elongation, approximately 20% and less than 35%, respectively (Figure 2a). The three samples of HE (HE1, HE2 and HE3) show extremely high %elongation (>350%) (Figure 2b,c). However, the three samples of LE (LE1, LE2 and LE3) show very low %elongation (<10%). Both HE and LE samples show only a small decrease in tensile strength and modulus at break to that of pure PLA. Typically, an increase in %elongation results in a noticeable reduction in both tensile strength and modulus at break. The appearance of films with low and high %elongation during tensile testing was also noted. It can be seen that the LE films became opaque with small stretch marks before fracture. Conversely, the HE films showed good strength with the formation of stress whitening and necking phenomena before fracture, which is possibly due to changes in molecular orientation during the stretching of polymer molecules. The tensile strength and %elongation of all compositions in Figure 1b are also shown in the Supporting Information Appendix A.

It should be noted that this result demonstrates an unusually large improvement in the tensile elongation at break (%elongation) of PLA blends that have not had any additional compatibilizer (e.g., tributyl citrate, acetyl triethyl citrate, dioctyl adipate) or copolymer added (e.g., poly(caprolactone-*co*-lactic acid, PLA–PCL diblock or triblock copolymers, poly(ethylene glycol)-*b*-PCL block copolymer) [25,27]. Generally, compatibilizers and copolymers lead to improvements in the interface between immiscible polymer blends, to increase flexibility, decrease glass transition temperature, reduce the tensile strength and result in an increase in %elongation of the PLA blends. To find out the reason behind the improved %elongation of this ternary-blended, melt-blown film of PLA/PCL/CAB, tensile-fractured surface morphology, miscibility, crystallinity, molecular weight, and chemical structures of the blends of both LE and HE samples were investigated.

### 3.3. Tensile-Fractured Surface Morphology by FE-SEM

The deformation of ternary-blended, melt-blown films of PLA/PCL/CAB from tensile-fractured surfaces of HE, LE and PLA films were observed by FE-SEM (Figure 3). For the PLA film (%elongation ≈ 20%), a smooth surface with fibrillated crazes through the sample was observed. The formed fine phase morphology was observed in HE1 and HE2 films, showing good homogeneity of polymer phases. For HE3 films, good interfacial interaction between the dispersed phase (PCL and CAB) and matrix (PLA) was also observed even though 15 wt% PCL and 25 wt% CAB were added. The formation of phase homogeneity and interfacial interactions in the HE1, HE2 and HE3 samples can lead to enhanced properties such as tensile elongation. However, LE1, LE2 and LE3 films with low %elongation (<10%) showed blends with coarse morphology and large dispersed phases, which may occur by the coalescence and aggregation of either PCL or CAB.

These differences in deformation and disintegration of the dispersed phase in the polymer blends are caused by the different compositions that effect viscosity, elasticity ratios and interfacial tension during the melt blending process [32]. Since all PLA/PCL/CAB blends experienced the same processing conditions, differential effects of the shear rate are not found. With the great improvement in %elongation and the formation of fine phase morphology of HE1, HE2 and HE3 films, these blended films were then termed “compatibilized blends”, which are generated by the reduction of interfacial adhesion between PLA with either PCL or CAB in the melt system.

### 3.4. Miscibility Observation by FT-IR and DSC

Miscibility is a vitally important property that can improve mechanical performance of polymer blends [31]. The blends are miscible or immiscible depending upon the competition between recrystallization and intermixing of polymers [17,19,30]. Miscibility promotes good interfacial adhesion of polymer molecules that may lead to improved polymer blends. FTIR spectroscopy was used to observe the molecular interactions in PLA/PCL/CAB blends (Figure 4). Pure PLA, PCL and CAB show FTIR absorption bands of –C–H stretching of –CH_2_ and –CH_3_ at 2700–2900 cm^−1^ and bands of –C=O stretching from ester or carboxylic acid groups at 1730–1750 cm^−1^. For the HE films, a shift to lower frequencies and broader peaks of –C=O stretching (ester or carboxylic acid groups) is observed (inset picture) when compared with pure PLA (sharper peak resulting from self-aggregation). This is possibly due to the diminished intermolecular packing (freedom movement) of polymer chains as well as possible molecular interactions from hydrogen bonding between CAB with either PLA or PCL, which cause a change in the electron cloud that alters the resonant frequency of that particular bond. This gives rise to different molecules having slightly different hydrogen bonding states leading to different frequencies and a broad band. For LE films (especially LE3), peaks of the –C=O stretching are simply the combination of the spectra of the two or three homopolymers, as expected for immiscible blends [33].

Miscibility observed by DSC was also studied and shown in Figure 5. The glass-transition temperature (T_g_) and melting temperature (T_m_) of melt-blown PLA films (from first heat runs, Figure 5a) are observed at approximately 64 °C and 148 °C, respectively. To subtract the effect of thermal history, the second heating runs (Figure 5c), including the cooling run (Figure 5b), were measured. The HE films display lower T_g_ and T_m_ than pure PLA films in both the first and second heating runs. There are no crystalline peaks in the cooling runs, only LE3 films show cold crystallization (at T_cc_) of PCL. For the second heating run, however, all HE films show the T_g_ peak of PLA, no T_m_ peaks of PCL, and very small crystalline peaks of T_m_ of PLA. Whereas, all LE films show both T_m_ crystalline peaks of PCL and PLA due to re-crystallization of their chains, which causes phase separation (as seen in Figure 3e,f,g) and immiscible blends. The reason for the peak shifting towards a lower temperature and the change in the DSC thermogram of ternary-blended, melt-blown films when PCL and CAB were added into the blends is attributed to the enhancement of chain mobility by the low molecular weight of PCL (no crystalline peak observed) as well as amorphous structures from CAB, resulting in HE films being miscible blends. These findings are similar to those seen in the work by El-Hadi et al., who reported that the addition of tributyl citrate (as plasticizer) and poly(3-hydroxy butyrate) to the PLA matrix leads to a reduction in T_g_, T_cc_ and T_m_ when compared to pure PLA [27].

### 3.5. Crystallinity by XRD

The XRD patterns of PLA and the ternary-blended, melt-blown films of PLA/PCL/CAB at different compositions are presented in Figure 6. The PLA film exhibited two broad peaks at 2𝜃 values of 14.5° and 30° and very small sharp peaks at 9.7° and 28.5°. All HE films showed similar spectra to pure PLA films but with lower peak intensities as lower amounts of PLA were present in the blends. No crystalline peaks associated with PCL were observed (Figure 5c). The LE films also showed broad peaks (PLA), albeit with lower intensities than HE films, alongside the crystalline peaks of PCL at 2𝜃 values at 20.8°, 21.5° and 23.7°, especially in LE3, which has the highest content of PCL. In addition, the degree of crystallinity of the samples according to DSC and XRD data was observed and showed similar values to the crystallinity of pure PLA film, which is approximately 25%.

### 3.6. Molecular Weight Analysis by GPC

The molecular weights of ternary-blended, melt-blown films of PLA/PCL/CAB were analyzed by GPC (Figure 7, which includes the raw materials PLA, PCL and CAB in Figure 7a). The weight-average molecular weights (*M_w_*) of all HE and LE films are lower than the raw materials. Interestingly, *M_w_* of all high %elongation films (HE1 (85/5/10), HE2 (75/10/15) and HE3 (60/15/25)) were observed to be lower than all low %elongation films (LE1 (85/10/5), LE2 (75/5/20) and LE3 (65/25/10)). The molecular weights of solvent-casted films (CF) (Figure 7c) and melt-extruded pellets before melt-blowing (EX) (Figure 7d) were also tested. For CF, similar GPC traces to those of the raw materials were observed. They showed two-separate peaks, one from PLA and CAB and another from PCL (especially when high PCL was added). For EX, the same trend in the GPC traces was observed as the ternary-blended, melt-blown films (Figure 7b) but with higher *M_w_*.

Comparison of *M_w_*, *M_n_* and PDI of HE/LE, EX and CF are shown in Figure 8. For EX and HE/LE samples, there is a decrease in *M_w_*, increase in *M_n_* and dramatic decrease in PDI of PLA/PCL/CAB blends when compared to CF. Comparing HE (HE1–85/5/10, HE2-75/10/15, HE3–60/15/25) and LE films (85/10/5, 75/5/20, 65/25/10), HE films show lower *M_w_* and *M_n_* values than LE films. This is potentially due to the melt process (in HE films) inducing molecular chain scission, which means that the thermal degradation may not be intensely promoted [29]. Moreover, the molecular chain scission may produce in situ generation of compatibilisers, which act in small quantities across the phase boundaries of PLA, PCL and CAB, and this seems to be a plausible concept in answering why HE films showed such improvement in tensile elongation at break. These in situ compatibilizers may not only be generated by chain scission (mostly from PCL) but also from chain-end reactions of hydroxyl groups and acyl ester groups from CAB.

### 3.7. Chemical Structure Observation by NMR

As aforementioned, miscibility, morphology, tensile elongation and molecular weight of PLA/PCL/CAB ternary blends were affected by the melt processes and the composition of PLA, PCL and CAB. The reduction in molecular weight suggests that molecular chain scission with in situ generation of compatibilisers might be occurring in the processing system. Therefore, chemical structures of HE and LE films were analyzed by ^1^H NMR (Figure 9a) and ^13^C NMR (Figure 9b) and compared to that of pure PLA, PCL and CAB (Figure 10).

From Figure 10, there are clear differences in the NMR peaks. The chemical shifts of pure PLA, PCL and CAB are marked with different symbols and colors to help identify the peaks in the ternary blends [34,35,36,37]. For ^1^H NMR; PLA (Figure 10a) shows peaks at chemical shifts of 5.13 and 1.56 ppm, which correspond to protons on methyne unit and methyl unit, respectively; PCL (Figure 10b) shows peaks at 4.07, 2.29 and 1.38 ppm, which correspond to protons on methylene units at different position on the backbone; CAB (Figure 10c) shows peaks at 2.33, 1.64 and 0.93 ppm of methylene (position a), methylene (position b) and methyl unit of the butyrate side group, at 2.11 ppm of methyl unit of acetate side group, and at 3.29–5.47 ppm of the cellulose backbone. For ^13^C NMR; PLA (Figure 10d) shows peaks at 16.7, 69.1 and 177.3 of methyl, methylene and carbonyl carbons, respectively; PCL (Figure 10e) shows peaks at 24.7–25.6, 28.5, 34.2 and 64.2 ppm of four methylene carbons (position a, b, c and d, respectively) and at 173.8 of carbonyl carbon; CAB (Figure 10f) shows peaks at 100.2, 73.0, 72.5, 76.8 and 71.9 ppm (potion a, b, c, d and e, respectively) of cellulose backbone, at 62.3 ppm of methylene carbon (position f), at 170.1 and 35.9 ppm of carbonyl and methyl carbon (position g and h) of the acetate side group, at 172.6, 21.0, 18.7 and 13.8 ppm of carbonyl carbon (position i), two methylene carbon (positions j and k) and methyl carbon (position l) of the butyrate side group.

The NMR spectra from the LE and HE films are shown in Figure 9a (^1^H NMR). All the peaks were identified with the relevant symbols of PLA, PCL and CAB, which were analyzed using the chemical shifts at the positions of hydrogen in each pure polymer (as seen in Figure 10). All ternary blend samples show identical peaks, except for the new quartet peaks at the chemical shifts between of 3.68–3.73 ppm, which is only seen in HE1, HE2 and HE3 (high %elongation films) (see zoomed in section). In Figure 9b (^13^C NMR), similar results from ^1^H NMR are seen with all peaks again being identified with the symbols of the chemical shifts from carbon atom positions of pure PLA, PCL and CAB. All films show identical peaks except for the peak at the chemical shift of 58.54 ppm, which is only seen in melted-blend films of HE1, HE2 and HE3 (see zoomed in section). The new peaks in the HE films from ^1^H NMR at 3.68–3.73 ppm and from ^13^C NMR at 58.54 ppm are thought to be from a methylene unit (–O–CH_2_–CH_3_) produced during the melt-blending process from mostly chain scission, but also including possible transesterification. In addition, to confirm this assumption, the solvent casted-films of PLA/PCL/CAB at the same compositions of HE1, HE2 and HE3 were also analyzed by NMR. The results show no extra peaks of ^1^H NMR such as the quartet peak at 3.68–3.73 ppm and ^13^C NMR peak at 58.54 ppm (see Appendix A). This is due to the solvent mixing process in the formation of solvent-mediated blends, which allows for only the intermingling between polymer chains and not for chain scission and the potential formation of new linkages between created molecules.

### 3.8. The Optimal Ratio Prediction Model and the Cost-Effectiveness Model

The optimal ratio prediction model consisted of two steps. The first step is to construct predictive models called “*the local models*”, $f_{\%\mathrm{CAB}}:\%\mathrm{PLA}\to \%E$, where each local model $f_{\%\mathrm{CAB}}$ represents a percentage value of CAB, where %CAB values are 5, 10, 15, 20, and 25% *w*/*w*, respectively. Each local model $f_{\%\mathrm{CAB}}$ was fitted from four points of %PLA experimental data, and all local models were fitted with 3-degree polynomial functions. The coefficients of the fitted polynomial functions and its residual (r^2^) are presented in Table 2.

Each local model $f_{\%\mathrm{CAB}}$ was considered for finding the maximization with respect to its related %PLA constraint. Table 3 shows the results of the local model maximizing, which predicts the highest value of %elongation and related %PLA ratio, respectively. All local models are shown as graphs in Figure 11.

For the second step, “*the global model*” was computed using the maximum value and its corresponding minimizer from the data in Table 3. The global model consists of 2 sub-models, $g_{1}:\%\mathrm{CAB}\to \%\mathrm{Elongation}$ and $g_{2}:\%\mathrm{CAB}\to \%\mathrm{PLA}$ where 5≤%CAB≤25. In the same manner as the first step, both two sub-models were fitted with a three-degree polynomial functions, as shown in Table 4, and plotted in Figure 12A,B, respectively.

Now, model $g_{1}$ is considered. Notice that the maximum value of this model $g_{1}$ is the highest predicted value of %elongation along with the corresponding optimal %CAB. In fact, by using a simple computation (or simple observation of Figure 12A), the maximum value of $g_{1}$ is 452 with 25 %CAB. This would satisfy the optimal requirements if only %elongation is of concern. However, in practice, other factors, such as cost, must always be considered so the cost-effectiveness model should also be analyzed. Based on the current information, the costs of raw materials are as following:
(i)USD 105/kg for CAB(ii)USD 10/kg for PCL(iii)USD 4.5/kg for PLA

This information suggests that the relation between %CAB and cost for each combination is:(2)$$\mathrm{Price}\left(x\right)=753.036+38.9816\left(x\right)+4.35016\left(x^{2}\right)-0.0887223(x^{3})$$
where x is %CAB. Subsequently, to calculate the trade-off between %elongation and cost of raw materials (with respect to %CAB) the maximizer of the function $g_{1}\left(x\right)/\mathrm{Price}\left(x\right)$ should be found as shown in Figure 13A. The calculations showed that 10.34 %CAB is the solution, therefore, using model $g_{2}$, we found that the corresponding %PLA for this solution is 83.60, as shown in Figure 13B.

Thus, the optimal ratio for the compound concerning the cost of raw materials is %PLA:%PCL:%CAB = 83.6:6.06:10.34. When using model $g_{1}$, the predicted output (%elongation) for this composition is 391.10%. However, in practical terms, we may approximate this composition to %PLA:%PCL:%CAB = 85:5:10 and its predicted output (%elongation) will be approximately 382%. From the experimental data, this ternary-blended, melt-blown films had a tensile elongation of 359 ± 6%.

## 4. Conclusions

Ternary-blended, melt-blown films of PLA/PCL/CAB were successfully fabricated with compositions selected from apparent miscibility using an optical ternary phase diagram as a guide. The films with high tensile elongation at break (%elongation) (HE1–85/5/10, HE2–75/10/15 and HE3–60/15/25) showed dramatic improvement in %elongation (>350%) compared to pure PLA (ca. 20%). The tensile-fractured surface morphology, miscibility, crystallinity, molecular weight determination and chemical structures of HE films revealed that the significant improvement in %elongation through melt-processing is mainly due to the thermally-induced molecular chain scission with potential in the in situ generation of compatibilizers, that act in small quantities across the phase boundaries of PLA, PCL and CAB. The range of compositions in which a significant increase in %elongation is observed is 55–85% *w*/*w* PLA, 5–20% *w*/*w* PCL and 10–25% *w*/*w* CAB. In terms of developing this research at scale, mathematic models were studied and showed that the optimal ratio for the cost effectiveness of blends (%elongation ≈ 390) was %PLA:%PCL:%CAB ≈ 85:5:10.

## Figures and Tables

**Figure 1 polymers-15-00303-f001:**
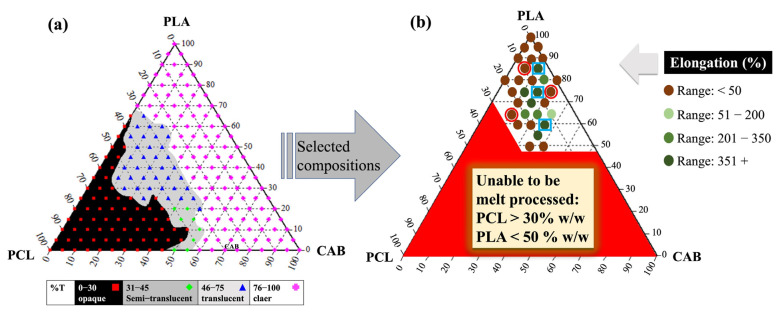
The optical ternary phase diagram of solvent-mediated blends of PLA/PCL/CAB constructed by the rapid scanning method (**a**), and the compositions that were able to be melt-blown into films with having different percentage of elongation at break (**b**). NB: Samples of HE (blue squares) and LE (red circles) were chosen to study further.

**Figure 2 polymers-15-00303-f002:**
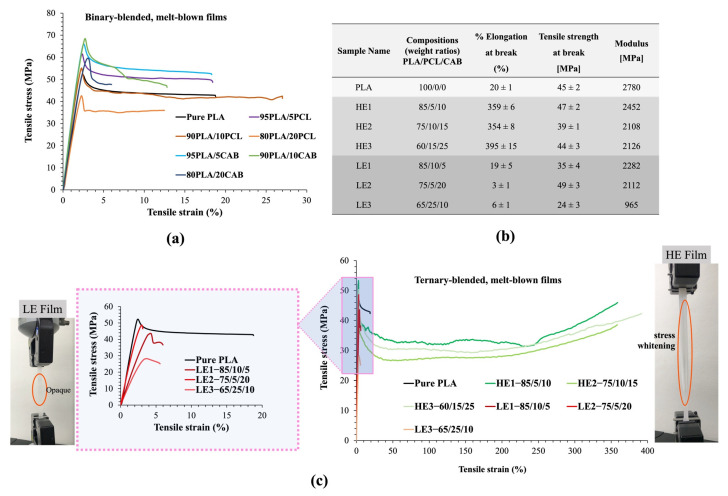
Tensile stress and tensile strain of binary-blended, melt-blown films (**a**), compositions and tensile property of ternary-blended, melt-blown films of LE and HE samples (**b**), tensile stress and tensile strain of ternary-blended, melt-blown films, inset figure shows the low tensile strain of LE film samples and appearances of LE and HE films from the tensile test (load cell of 1 kN and extension rate of 20 mm/min) before break (**c**). Note: Six samples were tested for each blended film.

**Figure 3 polymers-15-00303-f003:**
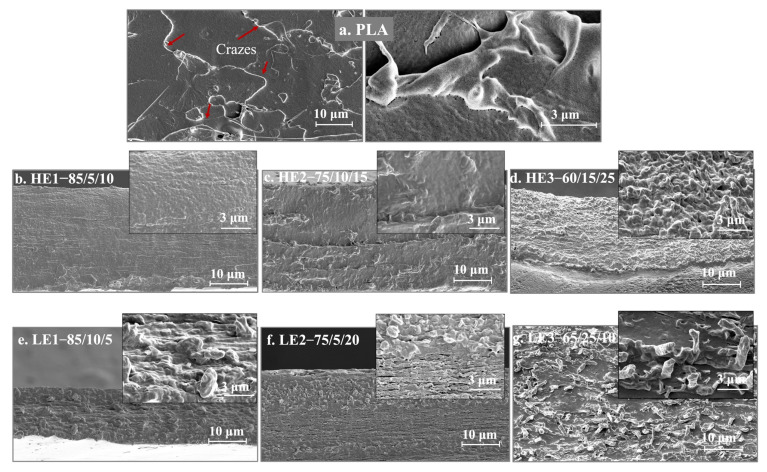
Morphology of ternary-blended, melt-blown films of PLA/PCL/CAB from tensile-fractured surface at 2000× and at 10,000× (inset images), together with PLA: (**a**) PLA, (**b**) HE1–85/5/10, (**c**) HE2–75/10/15, (**d**) HE3–60/15/25, (**e**) LE1–85/10/5, (**f**) LE2–75/5/20, (**g**) LE3–65/25/10.

**Figure 4 polymers-15-00303-f004:**
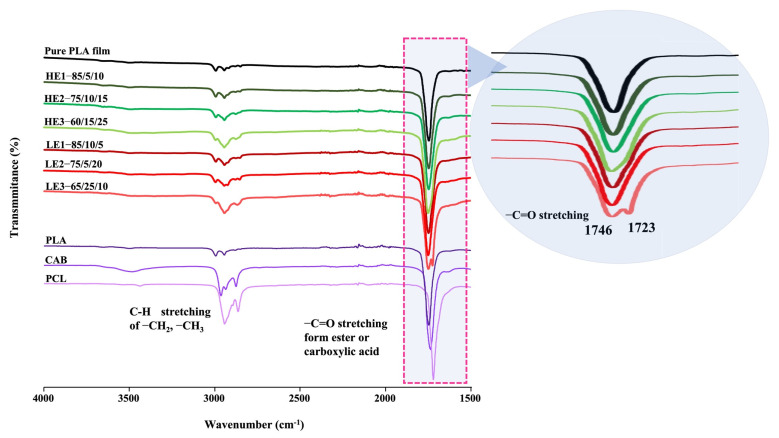
FTIR spectra of ternary-blended, melt-blown films of PLA/PCL/CAB at different compositions.

**Figure 5 polymers-15-00303-f005:**
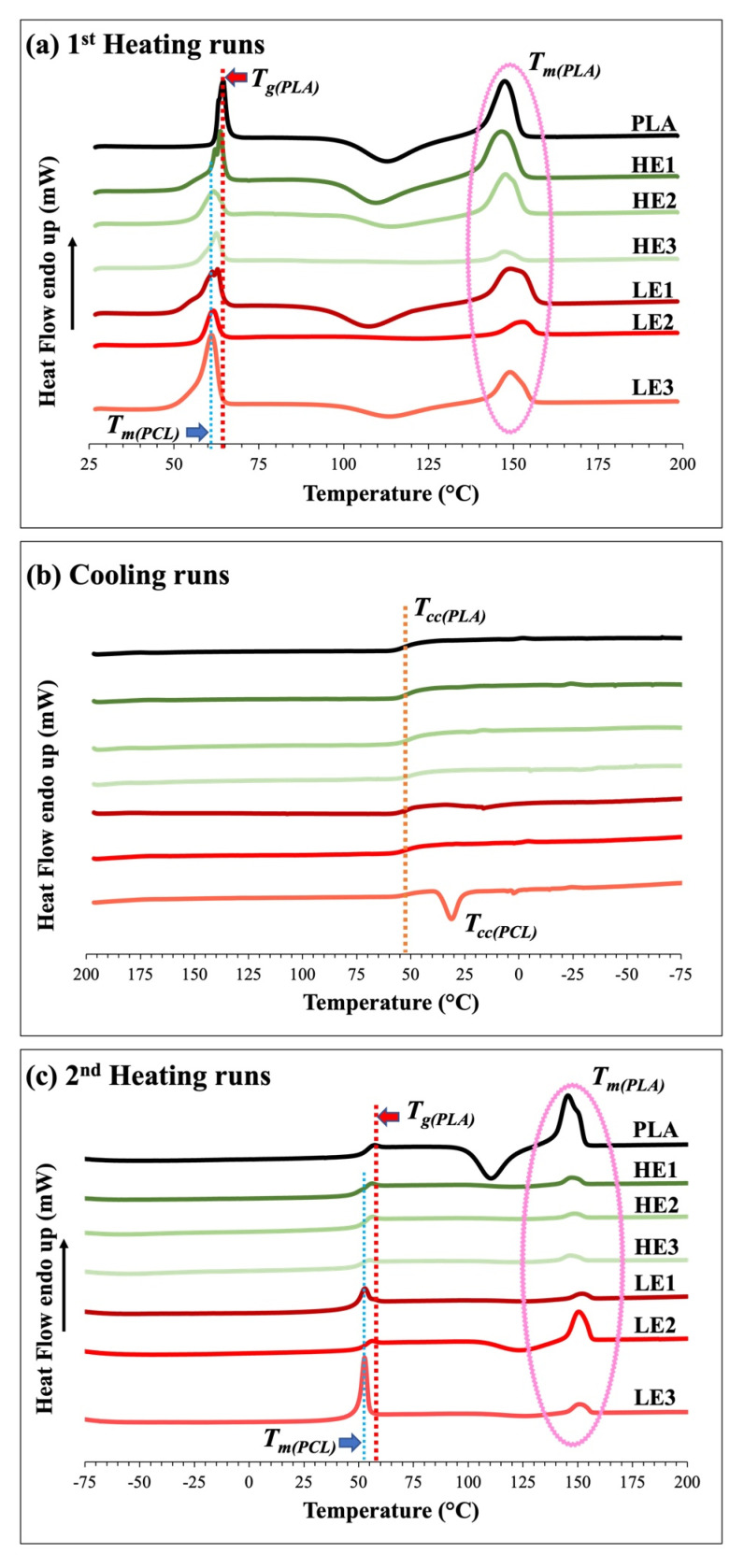
DSC thermograms of ternary-blended, melt-blown films of PLA/PCL/CAB; (**a**) first heating runs, (**b**) cooling runs and (**c**) second heating run.

**Figure 6 polymers-15-00303-f006:**
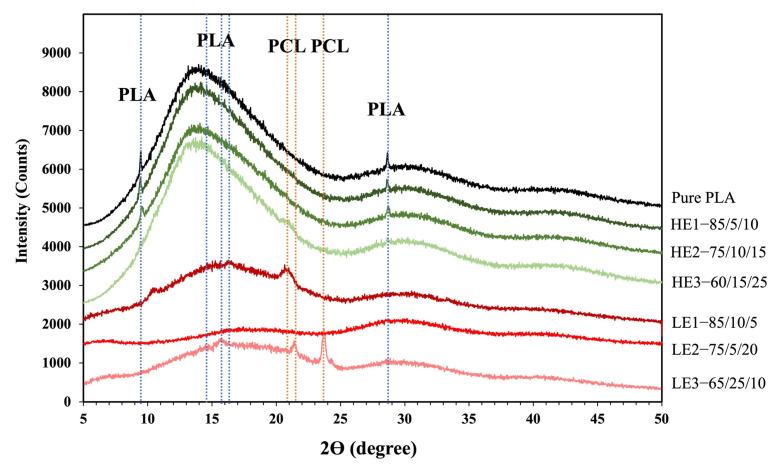
X-ray diffraction patterns of ternary-blended, melt-blown films of PLA/PCL/CAB at different compositions.

**Figure 7 polymers-15-00303-f007:**
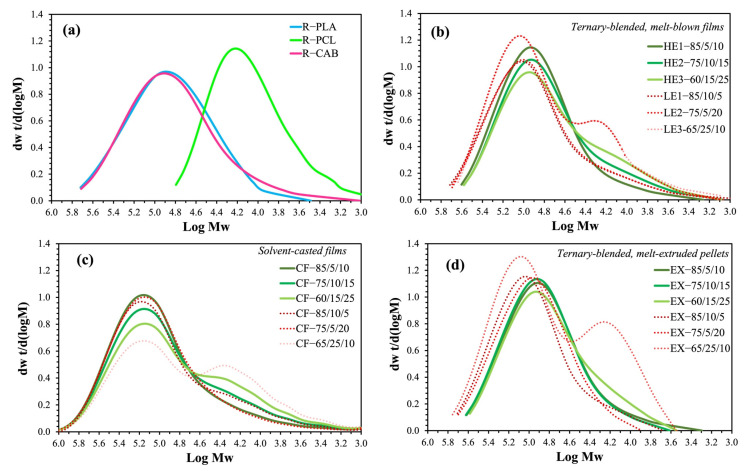
GPC traces of: (**a**) PLA, PCL and CAB pellets from raw materials, (**b**) ternary-blended, meth-blown films of PLA/PCL/CAB (HE and LE films), (**c**) solvent-casted films of PLA/PCL/CAB and (**d**) ternary-blended, melt-extruded pellets.

**Figure 8 polymers-15-00303-f008:**
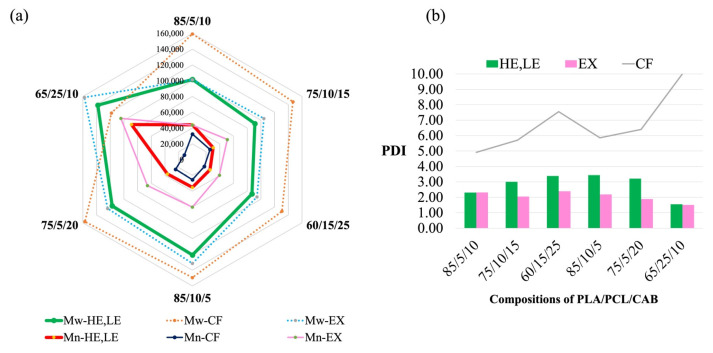
Molecular weights (*M_w_* and *M_n_*) of ternary-blended, melt-blown films (HE, LE), ternary-blended, melt-extruded pellets (EX) and solvent-casted films (CF) of PLA/PCL/CAB at different compositions (**a**), and their polydispersity (PDI) (**b**).

**Figure 9 polymers-15-00303-f009:**
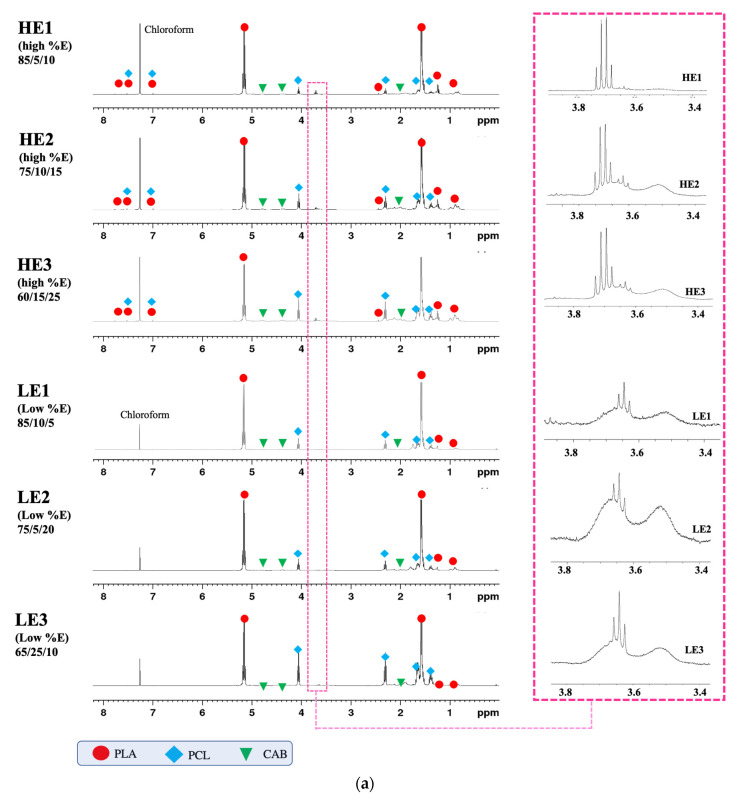
(**a**). ^1^H NMR of ternary-blended, melt-blown films of HE and LE films, showing zoom peaks at chemical shift of 3.4–3.8 ppm. (**b**). ^13^C NMR of ternary-blended, melt-blown films of HE and LE films, showing zoom peaks at the chemical shift of 58.4–58.8 ppm.

**Figure 10 polymers-15-00303-f010:**
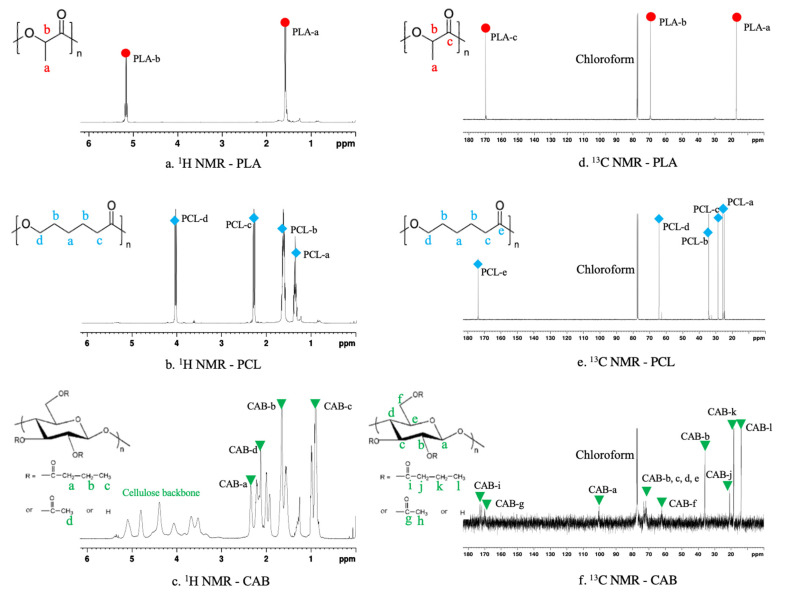
^1^H NMR (**a**–**c**) and ^13^C NMR (**d**–**f**) of PLA, PCL and CAB.

**Figure 11 polymers-15-00303-f011:**
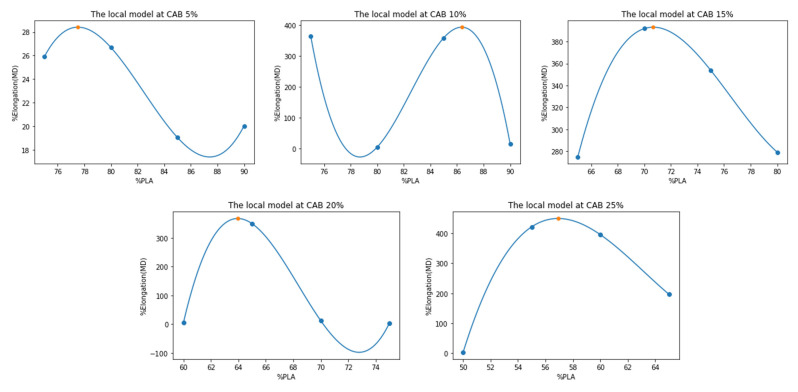
The local models at each %CAB with the maximum value highlighted (orange dots).

**Figure 12 polymers-15-00303-f012:**
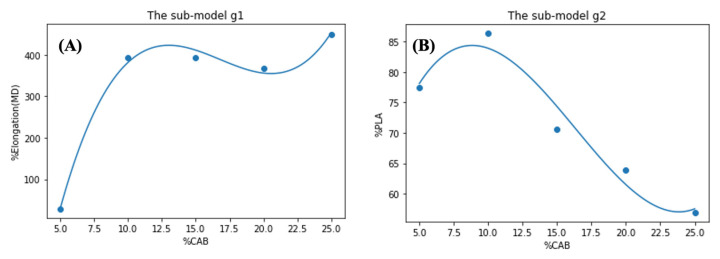
The global models of (%CAB, %Elongation (**A**)) and (%CAB, %PLA (**B**)).

**Figure 13 polymers-15-00303-f013:**
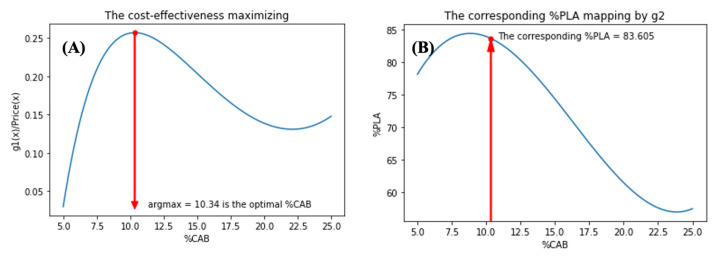
Models showing; the cost-effectiveness maximizing (**A**) and the corresponding %PLA mapping by *g*_2_ (**B**).

**Table 1 polymers-15-00303-t001:** Miscibility prediction of polymer pairs of PLA/PCL, PLA/CAB and PCL/CAB using critical solubility parameter differences [(∆δ)°^Crit^] from the Coleman and Painter approach, showing molar attraction constants (F) and molar volumes (V) together with derived critical solubility parameter differences (∆δ) [30].

Polymers		PLA	PCL	CAB
*F* (cal cm^3^)^1/2^		598	1017	1848
*V* (cm^3^ mol^−1^		49.5	98.3	154
δ (cal cm^−3^)^1/2^		12.1	10.3	12.0
Polymer pairs	Δδ	Interaction types	Δδ^°Crit^	Miscibility
PLA/PCL	1.8	Dispersive forces only	≤0.1	No
		Dipole-Dipole	0.5	No
		Weak	1.0	No
		Weak to moderate	1.5	No
PLA/CAB	0.1	Dispersive forces only	≤0.1	Yes
		Dipole-Dipole	0.5	Yes
		Weak	1.0	Yes
		Weak to moderate	1.5	Yes
		Moderate	2.0	Yes
		Moderate to strong	2.5	Yes
PCL/CAB	1.7	Weak to moderate	1.5	No
		Moderate	2.0	Yes
		Moderate to strong	2.5	Yes

**Table 2 polymers-15-00303-t002:** The local models of each %CAB.

$f_{\%CAB}\left(x\right)$	Coefficients				r^2^
	$x^{3}$	$x^{2}$	$x^{1}$	$x^{0}$	
$f_{5\%}\left(x\right)$	2.272000 × 10^2^	−5.621000 × 10^0^	4.618810 × 10^2^	−1.258204 × 10^4^	1.00
$f_{10\%}\left(x\right)$	−1.88362667 × 10^0^	4.66343800 × 10^2^	−3.84028203 × 10^4^	1.05204673 × 10^6^	1.00
$f_{15\%}\left(x\right)$	1.58053333 × 10^−1^	−3.62924000 × 10^1^	2.76149467× 10^3^	−6.92923100 × 10^4^	1.00
$f_{20\%}\left(x\right)$	1.34602667 × 10^0^	−2.76103200 × 10^2^	1.87995793 × 10^4^	−4.24739720 × 10^5^	1.00
$f_{25\%}\left(x\right)$	3.626400 × 10^−1^	−6.873060 × 10^2^	4.299589 × 10^3^	−8.847985 × 10^4^	1.00

**Table 3 polymers-15-00303-t003:** The local models maximizing.

Local Model $f_{\%CAB}$	$Max\;f\left(x\right)\;\%\mathrm{Elongation}\;(\mathrm{MD})$	$Argmax\;f\left(x\right)\;\%\mathrm{PLA}\;\mathrm{Ratio}$
$f_{5\%}$	28.411	77.518
$f_{10\%}$	393.439	86.349
$f_{15\%}$	393.147	70.670
$f_{20\%}$	367.503	63.953
$f_{25\%}$	449.434	56.923

**Table 4 polymers-15-00303-t004:** The two sub-models of the global model.

Global Model	Coefficients				r^2^
	$x^{3}$	$x^{2}$	$x^{1}$	$x^{0}$	
$g_{1}\left(x\right)$	3.15263333 × 10^−1^	−1.58769814 × 10^1^	2.53031510 × 10^2^	−8.76272200× 10^2^	0.994
$g_{2}\left(x\right)$	1.61313333 × 10^−2^	−7.90938571 × 10^−1^	1.01966238 × 10^1^	4.49026000× 10^1^	0.950

## Data Availability

The raw/processed data required to reproduce these findings cannot be shared at this time as the data also forms part of an ongoing study.

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
