# Peer review of "In Situ Compatibilized Blends of PLA/PCL/CAB Melt-Blown Films with High Elongation: Investigation of Miscibility, Morphology, Crystallinity and Modelling"

_polymers, 2023, doi:10.3390/polym15020303_

Round 1

Reviewer 1 Report

The article by Tuancharoensri et al is devoted to the search for the composition of the ternary polymer composition PLA, PCL, and cellulose acetate butyrate, which would provide the maximum elongation at break of the polymer film based on this system. This work is aimed at finding ways to improve the mechanical properties of polylactide, which is a biodegradable alternative to polyolefins in many applications, including as a packaging film material. The article presents a fairly detailed experimental study of compositions, including the construction of a simplified phase diagram based on transparency characterization, thermomechanical analysis, GPC, XRD, IR and NMR spectroscopy, and the construction of optimization models taking into account the cost of the initial components. The article combines elements of fundamental and applied research. The result obtained in it, i.e., an increase in the elongation at break parameter by an order of magnitude, is quite interesting, may well be due to reactive compatibilization, and deserves publication, taking into account a number of comments listed below.

1. Among the possible reasons for the qualitative change in the MWD and IR spectra for samples of the HE series, the authors indicate transesterification or thermal degradation reactions. In the first case, it is worth presenting a possible reaction scheme and discussing it on the basis of known literature data on mixtures of polylactide with other polymers. To check the possibility of thermal decomposition, it is appropriate to process the pure components of the mixture under the same conditions as for the compositions, followed by GPC and/or IR spectroscopy or TGA examination.

2. Approximation of experimental data by polynomials looks pointless, since there are no independent experiments that could confirm the statistical significance of the hypotheses made. The curves in fig. 11 change too abruptly to be physically justified.

3. From a practical point of view, it is more useful to give as a conclusion not the only optimal composition of the system, but the range of compositions in which a significant increase in elongation at break is observed. If this cannot be done based on the type of diagram in Fig. 1, it is worth exploring the most interesting area of this diagram in more detail.

4. Since the article has an applied focus, it is worth discussing what practical significance such a pronounced increase in elongation at break can have with a much lower variability in tensile strength and other parameters of the studied compositions, and also why this particular ternary system was chosen for research.

Minor comments

1. Please specify the data source in Table 1.

2. It is necessary to mark the curves in Figure 5b

3. What can be said about the degree of crystallinity of the samples according to DSC and XRD data?

4. The assignment of NMR signals of pure components of the system was not done for the first time and requires references or can be generally transferred to supplementary materials.

5. Two identical files are presented as supplementary materials.

Reviewer 2 Report

What are the purpose of processing ternary PLA/PCL/CAB and binary PLA/PCL and PLA/CAB blends? Do you have a specific application in mind?

The abstract should appeal to the reader. Novelty should be presented transparently. Research achievements should be mentioned quantitatively and qualitatively.

Usually, the elastic modulus is denoted by E, not the elongation.

The title is incomplete and phrases such as investigation or characterization should be added to the second part.

What is the reason for using the compatibilizer phrase in the title and keywords? While the abstract is not used.

The manner of referencing in the introduction should be changed and modified. In the first paragraph, which is general, 23 references are used. At least 15 of them are extra.

One of the best options to improve the properties of PLA, especially its brittleness, is the use of TPU, which has not been mentioned. Use these papers (3D printing of PLA-TPU with different component ratios: Fracture toughness, mechanical properties, and morphology).

How are the composition and melt process parameters selected?

How has the used model been verified?

The introduction should be revised. The introduction is written very superficially and briefly. To improve it, use up-to-date and suggested resources. (4D Printing-Encapsulated Polycaprolactone–Thermoplastic Polyurethane with High Shape Memory Performances, Experimental investigation on mechanical characterization of 3D printed PLA produced by fused deposition modeling (FDM), A New Strategy for Achieving Shape Memory Effects in 4D Printed Two-Layer Composite Structures).

Lines 180-192 should be removed or at least used elsewhere. This section cannot be included in the results.

Many parts of the results are reported, and analysis is not observed. It is better to use related sources to deepen the discussion and analysis. For example, use this source (Development of Pure Poly Vinyl Chloride (PVC) with Excellent 3D Printability and Macro- and Micro-Structural Properties) in the mechanical properties section.

Section 3.1 can also be used elsewhere and before presenting the results. It is suggested that this section be presented as an extended design to continue the research method.

Figure 2 has 4 sections but has 3 captions.

The conclusion should be modified like the abstract.

Round 2

Reviewer 1 Report

I am only partially satisfied with the revision since (1) no provisional reaction scheme was presented and (2) the fitting model was not verified from the viewpoint of its statistical significance. At the same time, I confirm my previous opinion that the obtained results are interesting and deserve publication. Moreover, I should note note that the presentation has been markedly improved. Thus, I support publication and encourage the authors to pursue further studies on the reactive compabilization of polymer blends.

Reviewer 2 Report

Accept.